# IgG Glycosylation Profiling of Peripheral Artery Diseases with Lectin Microarray

**DOI:** 10.3390/jcm11195727

**Published:** 2022-09-27

**Authors:** Siting Li, Jingjing Meng, Fang Xu, Qian Wang, Xinping Tian, Mengtao Li, Xiaofeng Zeng, Chaojun Hu, Yuehong Zheng

**Affiliations:** 1Department of Vascular Surgery, Peking Union Medical College Hospital, Chinese Academy of Medical Sciences and Peking Union Medical College, Beijing 100730, China; 2Department of State Key Laboratory of Complex Severe and Rare Diseases, Peking Union Medical College Hospital, Chinese Academy of Medical Science and Peking Union Medical College, Beijing 100730, China; 3Department of Rheumatology, Peking Union Medical College Hospital, Peking Union Medical College & Chinese Academy of Medical Sciences, National Clinical Research Center for Dermatologic and Immunologic Diseases (NCRC-DID), Key Laboratory of Rheumatology & Clinical Immunology, Ministry of Education, Beijing 100730, China; 4Department of Clinical Laboratory, Fifth Affiliated Hospital of Zhengzhou University, Zhengzhou 450052, China

**Keywords:** glycosylation, peripheral artery disease, lectin microarray, biomarker

## Abstract

Background: Inflammation plays a key role in the progression of atherosclerotic plaque for peripheral artery disease (PAD). Immunoglobulin G (IgG) glycosylation could modulate immunological effector functions and has been explored as biomarkers for various diseases. Methods: Lectin microarray was applied to analyze the expression profile of serum IgG glycosylation in patients with lower-extremity peripheral artery disease (LEPAD), carotid artery stenosis (CAS), abdominal aortic aneurysm (AAA), and healthy controls. Lectin blot was performed to validate the differences. Results: SNA (Sambucus nigra agglutinin) binding (preferred sialic acid) was significantly higher in the LEPAD (3.21 ± 2.06) and AAA (3.34 ± 2.42) groups compared to the CAS (2.47 ± 1.45) group. Significantly higher binding levels of ConA (Concanavalin A) (preferred mannose) and PSA (Pisum sativum agglutinin) (preferred fucose) were also observed in LEPAD compared to CAS patients. Among LEPAD patients, a significant lower binding level of Black bean crude (preferred GalNAc) was present for dyslipidemia patients. A higher binding level of MNA-M (Morniga M agglutinin) (preferred Mannose) and Jacalin-AIA (Artocarpus integrifolia agglutinin) (preferred Galβ3GalNAc) was observed for Fontaine severe patients. Higher binding levels of PHA-E (Phaseolus vulgaris Erythroagglutinin) and PHA-L (Phaseolus vulgaris Leucoagglutinin) (preferred Galβ4GlcNAc) were observed for diabetic patients, and higher binding of ASA (Allium sativum agglutinin) (preferred Mannose) was present in patients with hypertension. The level of high-sensitivity C-reactive protein (hsCRP) was positively associated with LTL (Lotus tetragonolobus lectin) (*r =* 0.44), PSA (*r =* 0.44), LCA (Lens Culinaris agglutinin) (*r* = 0.39), SNA (*r =* 0.57), and CSA (Cytisus sscoparius agglutinin) (*r =* 0.56). For CAS, symptomatic patients had lower binding levels of AAL (Aleuria aurantia lectin) (preferred fucose) and IAA (Iberis amara agglutinin) (preferred GalNAc). Blood total cholesterol level was positively associated with SNA-I (*r =* 0.36) and SBA (Soybean agglutinin) (*r = r =* 0.35). Creatinine levels were positively associated with lectins including, but not limited to, MNA-M (*r =* 0.42), CSA (*r =* 0.45), GHA (Glechoma hederacea agglutinin) (*r =* 0.42), and MNA-G (Morniga G agglutinin) (*r =* 0.45). Conclusion: LEPAD patients had increased IgG binding levels of SNA and ConA compared to CAS, which could provide potential diagnostic value. Fontaine severity was associated with Mannose-rich IgG N-glycan, while diabetic LEPAD correlated with bisecting GlcNAc. The levels of hsCRP and creatinine were positively associated with IgG fucosylation and galactosylation. Changes in IgG glycosylation may play important roles in PAD pathogenesis and progression.

## 1. Introduction

Peripheral artery disease (PAD) is a blood circulation disorder characterized by stenosis or occlusion of arteries other than the coronary system [1]. PAD is predominantly atherosclerotic and exhibits a higher risk of ischemic events and death compared to other cardiovascular pathologies [2]. Lower-extremity peripheral artery disease (LEPAD), the most prevalent PAD, affects >230 million people worldwide [3]. Carotid artery stenosis (CAS) is present in around 12.5% of males >70 years of age and significantly increases the risk of ischemic stroke [3,4].

The classic symptom of LEPAD is intermittent claudication (IC), while buttocks and thigh symptoms could also occur if proximal arteries were involved. If not properly treated, LEPAD could advance into critical limb-threatening ischemia (CLTI, Fontaine III, and IV) that may result in amputation or even death [3]. CAS, on the other hand, is associated with ischemic stroke, transient ischemic attack, or other neurologic symptoms [5]. Traditional diagnostic examinations include ankle brachial index (ABI) and color Doppler ultrasound. However, these tests were not routinely performed. Both LEPAD and CAS could be asymptomatic at an early stage, and some asymptomatic patients could still acquire CLTI or severe cerebrovascular events [4]. Specific serum biomarkers are urgently required for the early identification and prevention of diseases [6].

Inflammation and vessel remodeling plays a key role in the progression of atherosclerotic plaque for PAD, which prompts the screening of circulatory biomarkers such as proteins and non-coding RNAs in various studies [7]. In recent years, the importance of autoimmunity in atherosclerosis has been receiving increased interest [8,9]. Particularly, Immunoglobulin G (IgG) could alter the compartment of atherosclerotic lesions by targeting potential autoantigen (i.e., oxLDL) complement activation [10]. A change of IgG glycosylation profile has been reported to be associated with cardiovascular disease risk factors in some large cohorts [11,12]. However, currently, no study has specifically examined the IgG glycosylation status in LEPAD and CAS.

Lectin microarray is an emerging technology in glycosylation profiling. It has the advantages of being simple, rapid, high-throughput, and high-sensitivity compared with conventional glycan analysis methods such as mass spectrometry and has been applied in diverse studies of glycosylation and biomarker identification, especially for autoimmune diseases and cancers [13,14]. In this study, we firstly used lectin microarray to explore the expression profile of serum IgG glycosylation in patients with LEPAD, CAS, abdominal aortic aneurysm (AAA), and healthy controls. Lectin blot was performed to validate the differences.

## 2. Methods

### 2.1. Patients and Samples

From 2019 to 2021, 53 LEPAD, 75 CAS, and 75 disease control patients with AAA from the Vascular Surgery ward of Peking Union Medical College Hospital (PUMCH) were recruited for the study. One hundred healthy controls from the health examination center matched with age and sex were also included. Serum samples were collected upon admission, allowed to clot at room temperature for 30 min, centrifuged for 5 min at 1000× *g*, and stored at −80 °C. The study was approved by the institutional review board, and written informed consent was obtained from all patients.

Patients’ demographic data, initial symptoms, as well as pre-operative laboratory results were recorded. Patients with LEPAD were grouped according to the Fontaine classification system, where stages III and IV were defined as severe. Patients with CAS were grouped into symptomatic (with symptoms of amaurosis fugax, transient ischemic attacks, or ischemic stroke ipsilateral to the lesion) and asymptomatic. AAA controls had aneurysm diameters more than 5.5 cm for men or 5 cm for women, fast-growing (more than 10 mm/year), or symptomatic lesions. Patients with overlapping AAA and PAD have been excluded based on pre-operative aortic CTA and carotid ultrasound results. All the groups were also divided into subgroups according to concomitant diabetes, dyslipidemia, and hypertension.

### 2.2. Lectin Microarray Analysis

An example of the experimental procedure is illustrated in Figure 1. In total, 303 serum samples were detected using a commercial lectin microarray (BCBIO Biotech, Guangzhou, China) with 56 lectins, which had been proven of its reliability and used in biomarker findings previously [15,16]. Briefly, lectin microarrays were taken out from −80 °C and warmed up at room temperature for half an hour, and incubated with a blocking buffer (3% BSA in PBS) at room temperature for 2 h afterward. After washing three times with Phosphate-buffered saline Tween, (PBST), 200 μL of 1:1000 diluted samples of serum were added and incubated with the microarrays at 4 °C overnight. The microarrays were washed three times with PBST and then incubated with 5 mL of 1:1000 diluted Cy3-labeled goat anti-human IgG antibody (Jackson Immuno Research Labs) in the dark at room temperature for 1 h. Finally, after three PBST washes, microarrays were rinsed with D.I. water and dried. Microarrays were scanned with the GenePix 4000B Microarray Scanner (Molecular Devices, Sunnyvale, CA, USA).

For lectin array assays, the median foreground and background fluorescent intensity for each spot on the arrays were acquired using GenePix Pro 6.0 (Axon Instruments, Sunnyvale, CA, USA) software. We calculated the signal-to-noise ratio (S/N) (the medium intensity of the spot foreground relative to the background) of each lectin spot. To prevent bias of the lectin microarray from the inter-array, we normalized the S/N data in terms of quality controls’ value between arrays.

### 2.3. Lectin Blot Verification

To validate the results of the differences in lectin microarray analysis, lectin blot was used to detect serum samples collected from a smaller cohort containing 7–12 of LEPAD, CAS, or AAA patients, respectively, that were randomly selected from the microarray cohort (blot details can be found in the Appendix A). The following rules were used to identify significant differences in the binding activity of lectins for intra-disease group analysis: (a) fold change[group1(S/N)/group2(S/N)] ≥1.75 or <0.571, (b) *p*-value < 0.05. For inter-group analysis, two lectins (Con A and SNA) with the most significant differences between disease groups were selected

First, serum samples were diluted by 1 × PBS, mixed with gel electrophoresis loading buffer (CWbiotech) to a final 1:100 ratio, and boiled for 10 min. Twenty microliters per sample were separated by 10% sodium dodecyl sulphate—polyacrylamide gel electrophoresis (SDS-PAGE) and electrotransferred onto polyvinylidene fluoride membranes (Millipore, Billerica, MA, USA). After washing two times, the membrane was incubated with 10× Carbo-Free Blocking Solution (1:10; Vector Laboratories Inc., Newark, CA, USA) at room temperature for 2 h. Then, the membranes were washed twice and incubated with 20 μg/mL of Cy3-labeled (1:1000; GE Healthcare, Chicago, IL. USA) lectins at 4 °C overnight in the dark. Finally, the washed and dried membranes were detected by a fluorescence signal system of Typhoon FLA 9500 (GE Healthcare).

### 2.4. Statistical Analysis

R (v 4.0.2) and GraphPad Prism 8.0 (GraphPad Software Inc., San Diego, CA, USA) were used for statistical analyses and plot drawing. For continuous variables, inter- or intra-group results were compared by Student *t*-test, Mann–Whitney U test, one-way analysis of variance (ANOVA) with Tukey’s HSD test, or Kruskal–Wallis test if appropriate. The χ2 test or Fisher’s exact test was used for the comparison of categorical variables. Pearson correlation coefficient was used to analyze the relationship between the lectin binding level and clinical indicators. A *p*-value less than 0.05 was considered statistically significant.

## 3. Results

### 3.1. Patient Characteristics 

Table 1 summarized the baseline information of all study subjects. There were 44 (83.0%) and 63 (84.0%) males in the LEPAD and CAS groups, with an average age of 68.7 and 65.8, respectively. LEPAD patients have a higher rate of concomitant hypertension, dyslipidemia, diabetes, prior coronary artery disease (CAD), and stroke. Both PADs and AAA patients had a higher level of HCY and blood uric acid compared to healthy controls. The mean hsCRP levels were 6.2 mg/L, 4.5 mg/L, and 6.1 mg/L for LEPAD, CAS, and AAA patients, respectively. Regarding medication, 43 (81.1%) and 74 (98.7%) LEPAD and CAS patients, respectively, were taking statins, and most CAS patients (97.3%) were receiving antiplatelet treatment.

Table 2 further showed the clinical characteristics of LEPAD patients. Among all 53 patients, 24 (45.3%) had received prior lower-extremity operation. The distribution of patients for the Fontaine classification was 9 (17%) in grade IIa, 21 (39.6%) in IIb, 10 (18.9%) in grade III, and 13 (24.5%) in grade IV. There were 40, 32, and 28 patients with hypertension, dyslipidemia, and diabetes, respectively. No significant difference was observed for any Fontaine grade among different subgroups of patients. 

### 3.2. Lectin Microarray Results

The results of all 56 lectins binding from the microarray and their cluster analysis are illustrated in Appendix A. The scaled results of each lectin could be directly visualized by the heatmap. Table 3 summarizes the positive results of the inter-disease comparison. No significant difference was observed between any disease group with the healthy controls. SNA binding (preferred sialic acid) was significantly higher in the LEPAD (3.21 ± 2.06) and AAA (3.34 ± 2.42) groups compared to the CAS (2.47 ± 1.45) group. Significantly higher binding levels of ConA (preferred mannose) and PSA (preferred fucose) were also observed in LEPAD compared to CAS patients. 

Figure 2 and Figure 3 demonstrate the positive results from inter-disease subgroup analysis for LEPAD and CAS patients, respectively. Among the LEPAD patients, a significant lower binding level of Black bean crude (preferred GalNAc) was present for dyslipidemia patients compared to those with normal blood lipoprotein levels. A higher binding level of MNA-M (preferred Mannose) and Jacalin-AIA (preferred Galβ3GalNAc) was observed for Fontaine severe patients. Higher binding levels of PHA-E and PHA-L (preferred Galβ4GlcNAc) were observed for diabetic patients, and higher binding of ASA (preferred Mannose) was present in patients with hypertension. Concerning CAS, symptomatic patients had lower binding levels of AAL (preferred Fucose) and IAA (preferred GalNAc). Patients with diabetes had a higher binding level of ABA (preferred Gal(β1,3)GalNAc), while those with hypertension had a higher binding level of ASA (preferred Mannose). However, since none of these lectins in the CAS inter-group analysis had a fold change ≥1.75 or <0.571, they were not selected for lectin blot analysis.

### 3.3. Lectin Blot Analysis for LEPAD Patients

Lectin blots were performed for selected significant lectins, with results shown in Figure 4 and details in the Appendix A. For each group (subgroup), 7–12 serum samples were randomly chosen according to the total number in each group. Significant higher levels of SNA and ConA for the LEPAD patients compared to CAS patients were confirmed. A higher level of Jacalin-AIA was also present for the Fontaine severe group compared to the not-severe group. For the rest of the selected lectins, although the results were not significant, the tendency of difference was observed similarly to that from the microarray.

### 3.4. Clinical Relevance of Selective Lectins 

Figure 5 and Figure 6 showed significant correlations between selective lectins from microarray and laboratory results for LEPAD and CAS patients. Non-significant Pearson correlation coefficients (−0.1 ≤ r ≤ 0.1) were in blank color (R-squared values can be viewed in Appendix A). For LEPAD patients, female gender had higher levels of several lectins including PHA-E (*r =* −0.52), PHA-L (*r =* −0.51), and SBA (*r =* −0.48, preferred GalNAc). Fontaine severity was positively associated with lectins including MPL (*r =* 0.41, preferred Galβ3GalNAc) and VVA-mannose (*r =* 0.36, preferred mannose). The level of hsCRP was positively associated with various lectins such as LTL (*r =* 0.44, preferred fucose), PSA (*r =* 0.44, preferred fucose), LCA (*r =* 0.39, preferred fucose), SNA (*r =* 0.57, preferred sialic acid), and CSA (*r =* 0.56, preferred GalNAc). For CAS patients, patients receiving prior carotid artery operation had a higher binding level of BPL (*r =* 0.34, preferred Galβ3GalNAc). Blood total cholesterol level was positively associated with SNA-I (*r =* 0.36, preferred sialic acid) and SBA (*r =* 0.35). In addition, creatinine levels were positively associated with a number of lectins including, but not limited to, MNA-M (*r =* 0.42), CSA (*r =* 0.45), GHA (*r =* 0.42, preferred galactose), MNA-G (*r =* 0.45, preferred galactose), Jacalin-AIA (*r =* 0.46), and MPL (*r =* 0.41).

## 4. Discussion

Glycosylation takes place in more than half of human proteins and has profound structural and functional effects on its conjugate [17]. IgG glycosylation is generally N-glycosylation linked to an Asn on the heavy chain constant crystallizable fragment (Fc), and its alteration could modulate the effector functions such as complement-dependent cytotoxicity (CDC), antibody-mediated cellular cytotoxicity (ADCC), and antibody-dependent cell-mediated phagocytosis (ADCP) [18]. Additionally, O-glycosylation could occur on the hinge region of IgG3, which may provide proteolytic resistance as well as enhance antigen-binding of the Fab fragment by maintaining flexibility [19]. In this study, a lectin microarray with 56 kinds of lectin was used to detect the structures of serum IgG oligosaccharides in PAD patients. The technique has the advantage of basing directly on the interaction of lectins with glycans and does not require prior liberation of glycans from the core protein that may destroy their native structure [20].

We found that serum IgG of LEPAD patients had a significantly higher binding level of SNA (preferred sialic acid), ConA (preferred mannose), and PSA (preferred fucose) compared to CAS, with results of SNA and ConA confirmed by lectin blot analysis. Sialylation of the IgG Fc domain has been found to impair the complement-dependent cytotoxicity (CDC) effect [21], a reduction of which has been observed during inflammation and autoimmune disease flares [22]. It has been reported that loss of sialic acid might contribute to ischemic stroke [23]. Concerning mannose, Man-rich IgG N-glycan has an increased affinity for MBL, which could initiate the lectin complement cascade [24]. Our results suggested that although LEPAD and CAS both affect the peripheral vessel beds and could be concomitant in some patients, their pathogenesis and progression might be influenced by distinct immunological pathways. Alteration of affinity for lectins could serve as specific disease biomarkers for PAD patients and provide additive value in diagnosis.

For LEPAD subgroup analysis, we identified that, in LEPAD patients, higher binding levels of MNA-M (preferred mannose) and Jacalin-AIA (preferred Galβ3GalNAc) were observed for Fontaine severe patients. Fontaine severity was also positively associated with MPL (preferred Galβ3GalNAc) and VVA-mannose (preferred mannose), which also indicated that an increase of Fc-mannose was correlated with the severity of ischemic status. Galβ3GalNAc is a part of O-glycosylation, whose function has not been fully elucidated due to its complicated structure and resistance to proteolytic digestion [25]. Nevertheless, its alteration has been reported in some diseases [26,27].

Higher binding levels of PHA-E and PHA-L (preferred Galβ4GlcNAc) were observed in diabetic patients. Bisecting GlcNAc was associated with a decrease in core fucose, which could significantly increase its affinity for the FcγRⅢa receptor and promote ADCC. Similar results were observed for T2DM patients in previous studies [28,29]. Interestingly, the level of hsCRP was positively associated with various lectins such as LTL (preferred fucose), PSA (preferred fucose), LCA (preferred fucose), and SNA (preferred sialic acid). CRP is a crucial marker for PAD evaluation and prognosis [30]. The binding of CRP with circulating IgG immune complex has been found to increase the exposure of fucosyl residues in systemic lupus erythematosus patients [31]. What is more, symptomatic CAS patients had lower binding levels of AAL (preferred fucose), which suggested that lectin binding could serve as potential biomarker of PAD disease activity.

For CAS subgroup analysis, blood total cholesterol and LDL levels were positively associated with SNA-I (preferred sialic acid) and SBA (preferred GalNAc). It has been demonstrated that dyslipidemia was negatively associated with IgG sialyation in healthy subjects [11]. Since a majority of CAS patients were under statins therapy on admission, the results should be cautiously interpreted. Another interesting finding was that creatinine levels were positively associated with several lectins including CSA (preferred GalNAc), GHA (preferred galactose), MNA-G (preferred galactose), and MPL (preferred Galβ3GalNAc). Fc galactosylation is crucial for the initiation of the anti-inflammatory signaling cascade through binding to the inhibitory receptor FcγRIIb [24]. Creatinine was hypothesized to alter anti-inflammatory responses by interfering with the NF-κB pathway activation [32]. Higher creatinine levels have been found to be correlated with a lower chance of carotid artery vasospasm during stenting [33]. These findings suggested that the creatinine level could be considered as a reflection of inflammation besides renal function.

Our study has some limitations. Despite its easy accessibility, serum sample may not completely reflect the local glycosylation changes at the disease lesions. Although lectin microarray could serve as a convenient tool for glycosylation study, the exact structure of glycosylation could not be completely clarified. Due to a restricted number of clinical samples, disease controls apart from AAA were not included, and lectin blots should be performed with more subjects to verify microarray results. A larger cohort is required to validate the findings for diagnostic purposes. In the future, other techniques including affinity chromatography and the MALDI-TOF MS technique would be combined to further investigate the role of glycosylation in PAD pathogenesis.

## 5. Conclusions

LEPAD patients had increased IgG binding levels of SNA and ConA compared to CAS, which might provide potential diagnostic value. Fontaine severity was associated with Man-rich IgG N-glycan, while diabetic LEPAD correlated with bisecting GlcNAc. The levels of hsCRP and creatinine were positively associated with IgG fucosylation and galactosylation. Changes in IgG glycosylation may play important roles in PAD pathogenesis.

## Figures and Tables

**Figure 1 jcm-11-05727-f001:**
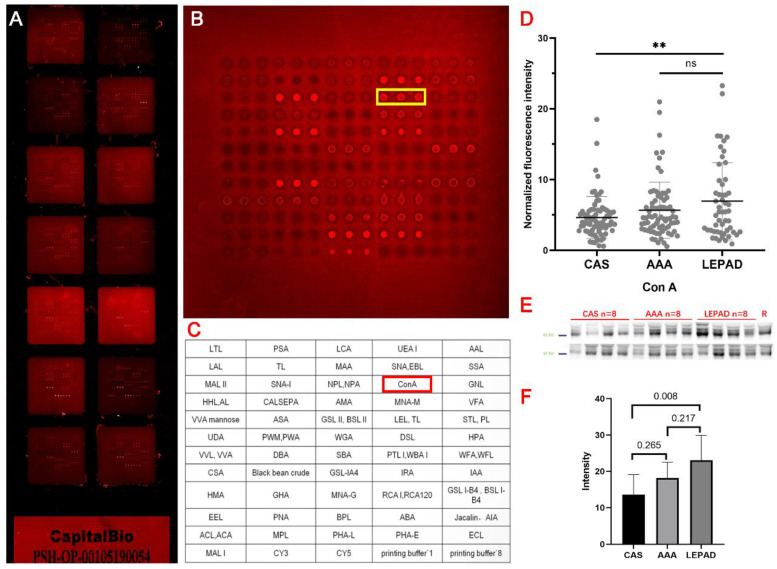
Example of lectin experimental procedure. (**A**) Each lectin microarray has 14 loading wells for 14 serum samples. (**B**,**C**) Each loading well contains 56 lectins and florescent or buffer controls with 3 iterations. Yellow box and red box indicate the position of lectin Con A. (**D**) Average normalized B/F signals of every lectin for each subject were calculated and compared within disease groups or subgroups. ** *p* < 0001; ns: *p* > 0.05. (**E**) A small cohort of serum was randomly selected for lectin blot verification. The IgG bands located at around 55kd on SDS-PAGE. (**F**). Lectin blot signal was analyzed and compared with lectin microarray results.

**Figure 2 jcm-11-05727-f002:**
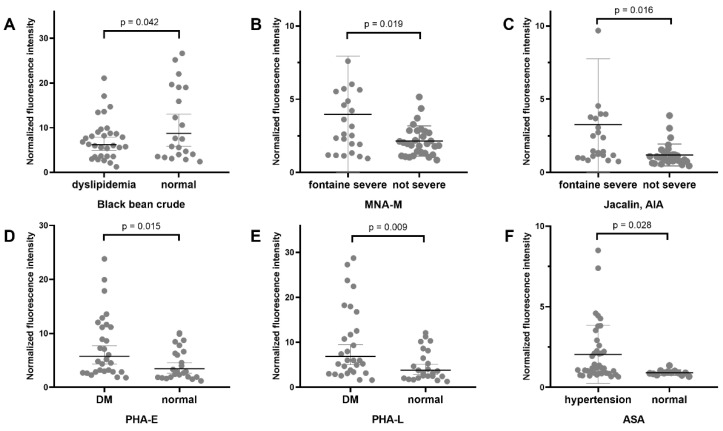
Positive results from lectin microarray for patients in different LEPAD subgroups. The following rules were used to identify significant differences for intra-disease group analysis: fold change[group1(S/N)/group2(S/N)] ≥1.75 or <0.571 and *p*-value < 0.05. DM, diabetes mellitus. Details of lectins and their binding glycan structures can be seen in the Appendix A. (**A**) Microarray result for black bean crude between LEPAD patients with or without dyslipidemia. (**B**) Microarray result for MNA-M between LEPAD patients with different Fontaine severity levels. (**C**) Microarray result for Jacalin between LEPAD patients with different Fontaine severity levels. (**D**) Microarray result for PHA-E between LEPAD patients with or without diabetes. (**E**) Microarray result for PHA-L between LEPAD patients with or without diabetes. (**F**) Microarray result for ASA between LEPAD patients with or without hypertension.

**Figure 3 jcm-11-05727-f003:**
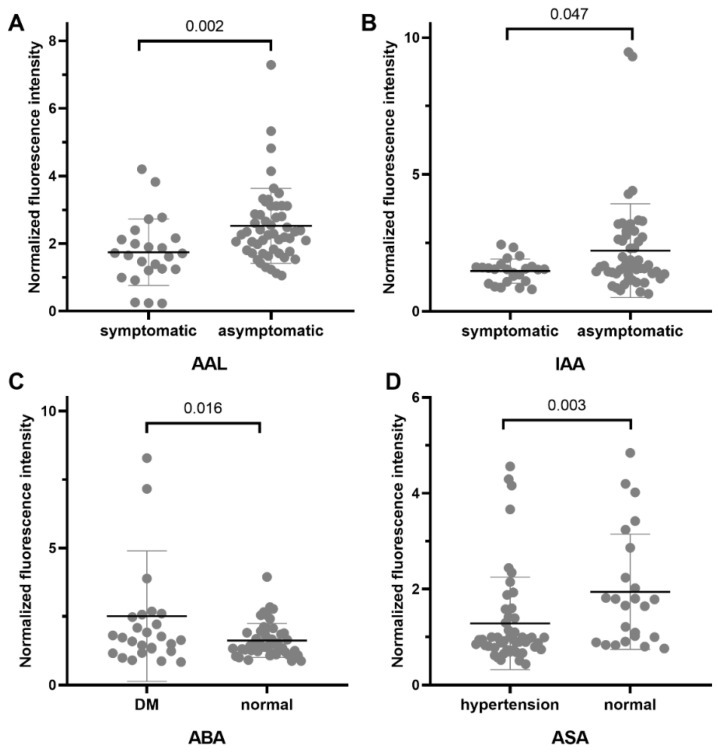
Positive results from lectin microarray for patients in different CAS subgroups. The following rules were used to identify significant differences for intra-disease group analysis: fold change[group1(S/N)/group2(S/N)] ≥1.75 or <0.571 and *p*-value < 0.05. DM, diabetes mellitus. Details of lectins and their binding glycan structures can be seen in the Appendix A. (**A**). Microarray result for AAL between symptomatic or asymptomatic CAS patients. (**B**). Microarray result for IAA between symptomatic or asymptomatic CAS patients. (**C**). Microarray result for ABA between CAS patients with or without diabetes (**D**). Microarray result for ASA between CAS patients with or without hyperntension.

**Figure 4 jcm-11-05727-f004:**
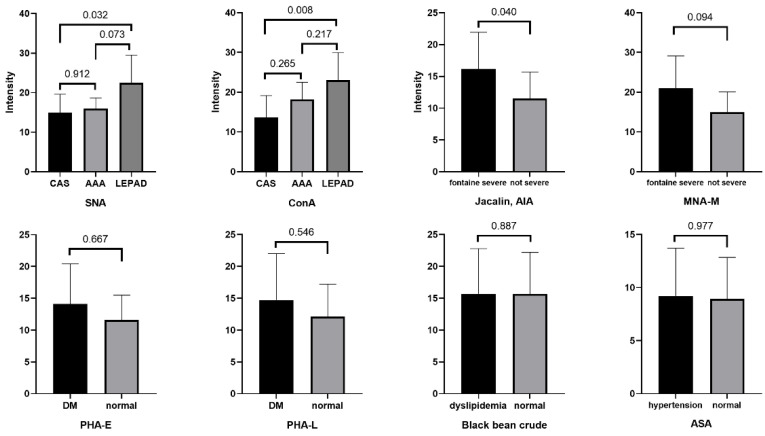
Lectin blot verification of selected lectins for LEPAD patients. For SNA and ConA, 7 and 8 samples in each disease group were included. For Jacalin-AIA and MAN-M, 9 samples in each subgroup were included. For Black bean crude and ASA, 12 samples in each subgroup were included. CAS, carotid artery stenosis; AAA, abdominal aortic aneurysm; LEPAD, lower-extremity peripheral artery disease; DM, diabetes mellitus. Details of lectins and their binding glycan structures can be seen in the Appendix A.

**Figure 5 jcm-11-05727-f005:**
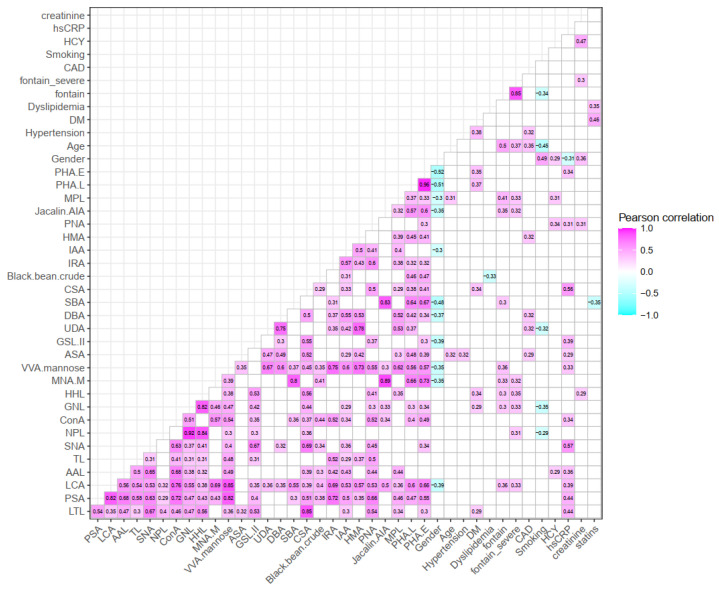
Heatmap of significant correlation between selective lectins and laboratory results for LEPAD patients. Non-significant Pearson correlation coefficients (−0.1 ≤ *r* ≤ 0.1) are in blank color.

**Figure 6 jcm-11-05727-f006:**
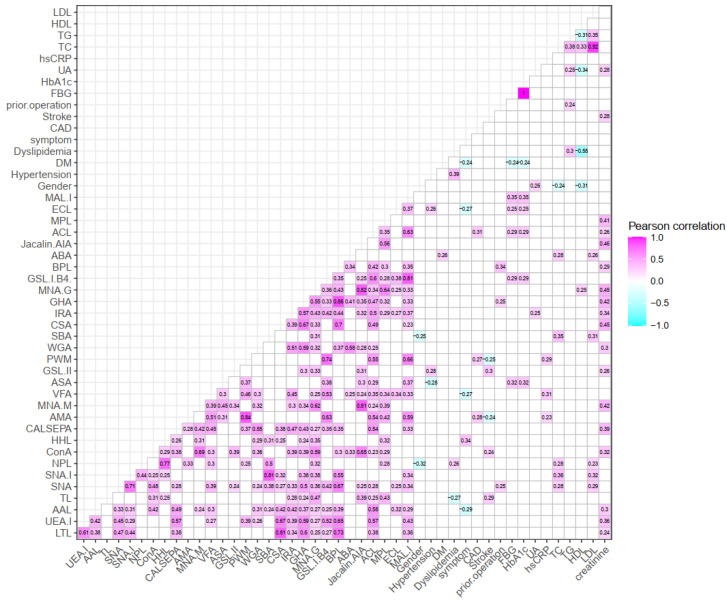
Heatmap of significant correlation between selective lectins and laboratory results for CAS patients. Non-significant Pearson correlation coefficients (−0.1 ≤ *r* ≤ 0.1) are in blank color.

**Table 1 jcm-11-05727-t001:** Clinical characteristics of patients.

No. (%) or Mean ± SD	LEPAD (*n* = 53)	CAS (*n* = 75)	AAA (*n* = 75)	HC (*n* = 100)
Male, n%	44 (83)	63 (84)	61 (81.3)	81 (81)
Age (years)	68.7 ± 9.2	65.8 ± 7.9	69.9 ± 8.6	66.5 ± 6.8
Hypertension, n%	40 (75.5)	51 (68)	60 (80)	59 (59)
Dyslipidemia, n%	31 (58.5)	37 (49.3)	42 (56)	48 (48)
Diabetes, n%	29 (54.7)	27 (36)	10 (13.3)	22 (22)
CAD, n%	19 (35.8)	19 (25.3)	29 (38.7)	NA
Stroke, n%	14 (26.4)	14 (18.7)	11 (14.7)	NA
Ever smoked, n%	31 (58.5)	32 (42.7)	42 (56)	NA
Total glycerol (mmol/L)	1.3 ± 0.7	1.2 ± 0.6	1.5 ± 0.8	1.5 ± 0.8
LDL cholesterol (mmol/L)	2.3 ± 0.9	2.0 ± 0.7	2.6 ± 1.0	3.0 ± 3.1
HDL cholesterol (mmol/L)	1.1 ± 0.4	1.1 ± 0.3	1.0 ± 0.2	1.4 ± 0.6
Total cholesterol (mmol/L)	3.9 ± 1.0	3.6 ± 0.9	4.2 ± 1.2	4.6 ± 1.1
HCY (μmol/L)	15.8 ± 4.7	15.5 ± 5.2	17.4 ± 8.2	14.4 ± 3.2
hsCRP (mg/L)	6.2 ± 14.9	4.5 ± 14.2	6.1 ± 19.3	1.6 ± 3.6
Blood creatinine (μmol/L)	84.8 ± 30.8	81.9 ± 26.5	89.2 ± 33.3	82.8 ± 21.4
Blood uric acid (μmol/L)	365.8 ± 115.1	360.2 ± 81.8	378.4 ± 80.9	353.6 ± 84.8
FBG (mmol/L)	6.4 ± 2.2	6.1 ± 1.4	5.4 ± 1.3	6.1 ± 1.7
HbA1c, %	6.8 ± 1.2	6.1 ± 1.4	6.1 ± 1.2	5.9 ± 0.9
Medication, n%				
Statin	43 (81.1)	74 (98.7)	36 (48)	NA
Antiplatelet	42 (50.6)	73 (97.3)	56 (74.7)	NA
Antihypertensive	42 (50.6)	42 (56)	37 (49.3)	NA

SD, standard deviation; LEPAD, Lower-extremity artery disease; CAS, carotid artery stenosis; AAA, abdominal aortic aneurysm; HC, health control; CAD, coronary artery disease; LDL, low-density lipoprotein; HDL, higher-density lipoprotein; HCY, homocysteine; CRP, C-reactive protein; FBG, fasting blood glucose; HbA1c, glycosylated hemoglobin.

**Table 2 jcm-11-05727-t002:** Clinical characteristics for lower-extremity artery disease patients.

No. (%)		All (*n* = 53)	Hypertension (*n* = 40)	Dyslipidemia (*n* = 31)	Diabetes (*n* = 29)	*p*
Prior lowerextremity operation		24 (45.3)	19 (47.5)	14 (45.2)	16 (55.2)	0.837
Fontaine classification						
Not severe	IIa	9 (17)	7 (17.5)	3 (9.7)	2 (6.9)	0.492
	IIb	21 (39.6)	16 (40)	13 (41.9)	13 (44.8)	0.979
all		30 (56.6)	23 (57.5)	16 (51.6)	15 (51.7)	0.935
Severe	III	10 (18.9)	8 (20)	7 (22.6)	7 (24.1)	0.955
	IV	13 (24.5)	9 (22.5)	8 (25.8)	7 (24.1)	0.993
all		23 (43.4)	17 (42.5)	15 (48.4)	14 (48.3)	0.935

**Table 3 jcm-11-05727-t003:** Positive results from lectin microarray inter-group analysis.

Lectin	Preferred Sugar	Normalized Fluorescence Intensity (Mean ± SD)					Fold Change		
		LEPAD	CAS	AAA	HC	LEPAD/CAS	LEPAD/AAA	CAS/AAA	LEPAD/HC
SNA	Sialic acid	3.21 ± 2.06	2.47 ± 1.45	3.34 ± 2.42	3.23 ± 3.21	1.301 *	0.963	0.740 *	0.996
ConA	Mannose	6.95 ± 5.43	4.64 ± 2.96	5.50 ± 3.77	5.97 ± 3.65	1.497 **	1.262	0.843	1.165
PSA	Fucose	3.62 ± 3.06	2.78 ± 1.39	2.99 ± 2.65	3.17 ± 1.76	1.302 *	1.215	0.931	1.142

** *p* < 0.01,* *p* < 0.05.

## Data Availability

The datasets generated during and/or analyzed during the current study are available from the corresponding author on reasonable request.

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
