# Peer review of "IgG Glycosylation Profiling of Peripheral Artery Diseases with Lectin Microarray"

_jcm, 2022, doi:10.3390/jcm11195727_

Round 1
Reviewer 1 Report
The article by Li and Meng (co-first authors) examines measures of glycolysation in the serum of patients vs. controls and draws the conclusion that some correlated with -- what the authors label as LEAD for lower extremity peripheral arterial disease. The abbreviation is not used in the way and should be discouraged. While I understand why the authors did this it is distracting to read.
I will start with the positive findings: the authors selected a relatively unexplored area and the methodology of the serum assays seemed well done. To this reviewer the single strongest positive from study was the approach of using risk factor controlled "control" subjects. This is not always done and should be better pointed out by the authors.
I offer the following suggestions to Editor and author:
1) The manuscript needs to be reviewed more carefully. For example AAA is used a name of a group of subjects without being previously defined; that is done in Table 1 but finding were already referred to in the abstract and manuscript. It (AAA) is a commonly used abbreviation but given the use of the abbreviation of LEAD (which is non commonly and probably should not be used). Diseases of AAA and PAD (and even CAS for cerebrovascular disease) overlap within individual patients and at least some details on how this was addressed would be valuable.
2) The reviewer appreciates that measures made in the plasma allow the findings to be generalizable but it raises the question as to the disease subtype and perhaps the specific question as to whether the target organ (the legs in patients with PAD, the aorta in AAA, and of course brain in CAS) play any role it what is measured. These are all diseases of
atherosclerosis, which is a generalized process.
3) Many of the finding of the differences between groups (for example LEAD vs. control) are small and unlikely to be useful at this stage. There can be differences between the groups but the overlap of values places the data at the point of not being useful for diagnostic purposes. Figure 1 being a clear example. Much of the tone of the findings have to be lowered.
4) The correlation coefficients, as presented, are a bit difficult to follow. The value to rsquared (not r) shows the percent to which one value drives the other. Something can be statistically significant but have a r (and thus r squared) that is so low as to be not likely to be important. Might the authors consider presenting all values where r squared is 25% (or some other number) putting the r at .5. Unless a strong statistical argument can be made I think this would be more useful.
5) In the lines around 50 - 54, this reviewer is not sure of the statement. PAD is caused by atherosclerosis and diagnosed by a low ankle-brachial blood pressure index. Patients may or may not have leg symptoms but one does not have to symptomatic intermittent claudication before having critical limb ischemia.
Author Response
Comments and Suggestions for Authors
Lectin microarray is an effective and reliable technique for analysing glycan structure. Recent studies have confirmed that IgG glycosylation abnormalities play a relevant role in the occurrence and development of different diseases, such as cancer or cardiovascular and autoimmune diseases. In this context, the manuscript elaborated by Siting Li et al. explores this relevant area of research.
To improve the quality of this research article we recommended:
- The Abstract is incomplete and sometimes challenging to understand (please revise the English line 21). The authors should also add the background of their research. All the acronyms should be explained in this part of the manuscript (e.g. AAA, SNA-I, or SBA).
Thank you for your valuable suggestion. We have revised the abstract accordingly (page 1, line 17).
- Please define the ’’smaller cohort of randomly-selected LEAD, CAS, or AAA patients’’ from which serum samples have been collected in Section 2.3.
This point has been further addressed in the manuscript (page 4, line 128).
- Kindly explain ’’HbA1c and SD’’ in the table 1 legend.
This point has been explained in the legend accordingly (page 5, line 171).
- Please revise the data presented in table 2. No SD is shown in this table (please see the first column - ’’No. (%) or mean ± SD’’).
The table has been revised accordingly (page 5, line 175).
- Where are found in the main text Figures S1 and S2? Please revise this aspect (see line 167).
Explanation of figure S1 and S2 have been added to the manuscript accordingly (page 5, line 176). Figure S1 and S2 could provide a direct overview of microarray results for all the lectins. Quantitative analysis of lectins with statistical significance were shown in Table 3 and Figure 2-6.
- Kindly explain all the acronyms used in the legend of Figures 2, 3 and 4.
All the acronyms have been explained accordingly. For the names of lectins, we have uploaded the names for each lectin in the supplementary file: “contents of 56 lectins and their binding glycan structures.xslx”.
- Please add a reference in line 48 according to the data presented.
The reference has been added accordingly (page 11, line 341).
Reviewer 2 Report
Lectin microarray is an effective and reliable technique for analysing glycan structure. Recent studies have confirmed that IgG glycosylation abnormalities play a relevant role in the occurrence and development of different diseases, such as cancer or cardiovascular and autoimmune diseases. In this context, the manuscript elaborated by Siting Li et al. explores this relevant area of research.
To improve the quality of this research article we recommended:
- The Abstract is incomplete and sometimes challenging to understand (please revise the English line 21). The authors should also add the background of their research. All the acronyms should be explained in this part of the manuscript (e.g. AAA, SNA-I, or SBA).
- Please define the ’’smaller cohort of randomly-selected LEAD, CAS, or AAA patients’’ from which serum samples have been collected in Section 2.3.
- Kindly explain ’’HbA1c and SD’’ in the table 1 legend.
- Please revise the data presented in table 2. No SD is shown in this table (please see the first column - ’’No. (%) or mean ± SD’’).
- Where are found in the main text Figures S1 and S2? Please revise this aspect (see line 167).
- Kindly explain all the acronyms used in the legend of Figures 2, 3 and 4.
- Please add a reference in line 48 according to the data presented.
- The references should be revised according to the journal recommendations.
Author Response
1) The manuscript needs to be reviewed more carefully. For example AAA is used a name of a group of subjects without being previously defined; that is done in Table 1 but finding were already referred to in the abstract and manuscript. It (AAA) is a commonly used abbreviation but given the use of the abbreviation of LEAD (which is non commonly and probably should not be used). Diseases of AAA and PAD (and even CAS for cerebrovascular disease) overlap within individual patients and at least some details on how this was addressed would be valuable.
Thank you for your advice. We have changed all the abbreviations for lower extremity peripheral artery disease into LEPAD. The definition of AAA has been added (page 2, line 96). Patients with overlapping AAA and PAD have been excluded based on aortic CTA and carotid ultrasound results. This point has been further addressed accordingly (page 2, line 97).
2) The reviewer appreciates that measures made in the plasma allow the findings to be generalizable but it raises the question as to the disease subtype and perhaps the specific question as to whether the target organ (the legs in patients with PAD, the aorta in AAA, and of course brain in CAS) play any role it what is measured. These are all diseases of
atherosclerosis, which is a generalized process.
Indeed, atherosclerosis plays an important role in the pathogenesis of all three groups. Ideally, it would be better to collect samples adjacent to the lesions in the target organ (such as the aneurysm wall), yet these tissues are hard to acquire in the endovascular era. Blood sample, on the other hand, are easily-accessed and has the potential for disease early diagnosis. We have added this point to the Discussion part accordingly (page 11, line 306). Additionally, before conducting the experiment, we assumed that the IgG glycosylation would be similar for LEPAD and CAS. Our results suggested that besides the generalized atherosclerotic process, the inflammatory status might be different for the two diseases.
3) Many of the finding of the differences between groups (for example LEAD vs. control) are small and unlikely to be useful at this stage. There can be differences between the groups but the overlap of values places the data at the point of not being useful for diagnostic purposes. Figure 1 being a clear example. Much of the tone of the findings have to be lowered.
Thank you for your valuable suggestion. We understand that as a pilot study, the results could not reach a very concrete conclusion. Thus, in the future, we planned to apply MALDI-TOF MS technique (a more economical approach) in a larger cohort to validate the differences from this study. This part has been addressed in the limitation section (page 11, line 311), and the conclusion was revised accordingly (page 11, line 315).
4) The correlation coefficients, as presented, are a bit difficult to follow. The value to rsquared (not r) shows the percent to which one value drives the other. Something can be statistically significant but have a r (and thus r squared) that is so low as to be not likely to be important. Might the authors consider presenting all values where r squared is 25% (or some other number) putting the r at .5. Unless a strong statistical argument can be made I think this would be more useful.
Thank you for your insightful advice. R squared could, indeed, show the percent to which one value drives the other. Nevertheless, we chose r in that it could directly visualize whether it’s a positive or negative correlation, and the absolute value of r could also reflect the strength of association (i.e., .1 to .3 as small, .3 to .5 as median, and .5 to 1.0 as strong). We have uploaded the pictures shown with r squared as additional supplementary files, and wish this could help present the results together with figure 5 and figure 6. The manuscript has been revised accordingly (page 8, line 228).
5) In the lines around 50 - 54, this reviewer is not sure of the statement. PAD is caused by atherosclerosis and diagnosed by a low ankle-brachial blood pressure index. Patients may or may not have leg symptoms but one does not have to symptomatic intermittent claudication before having critical limb ischemia.
Thank you for pointing out the issue. We strongly agreed that one does not have to symptomatic intermittent claudication before having critical limb ischemia. Thus, early screening and diagnosis of the disease is important. The manuscript has been revised accordingly (page 4, line 128).
Round 2
Reviewer 2 Report
The quality of the manuscript has greatly improved with more details and clarifications.
I recommend the paper for publication in the present form.